https://doi.org/10.1038/s42004-021-00452-y　　OPEN
# Silver-assisted gold-catalyzed formal synthesis of the anticoagulant Fondaparinux pentasaccharide

Gulab Walke[1], Niteshlal Kasdekar[1], Yogesh Sutar[1] & Srinivas Hotha [1✉]

Clinically approved anti-coagulant Fondaparinux is safe since it has zero contamination problems often associated with animal based heparins. Fondaparinux is a synthetic penta-saccharide based on the antithrombin-binding domain of Heparin sulfate and contains glu-cosamine, glucuronic acid and iduronic acid in its sequence. Here, we show the formal synthesis of Fondaparinux pentasaccharide by performing all glycosidations in a catalytic fashion for the first time to the best of our knowledge. Designer monosaccharides were synthesized avoiding harsh reaction conditions or reagents. Further, those were subjected to reciprocal donor-acceptor selectivity studies to guide [Au]/[Ag]-catalytic glycosidations for assembling the pentasaccharide in a highly convergent [3 + 2] or [3 + 1 + 1] manner. Cat-alytic and mild activation during glycosidations that produce desired glycosides exclusively, scalable route to the synthesis of unnatural and expensive iduronic acid, minimal number of steps and facile purifications, shared use of functionalized building blocks and excellent process efficiency are the salient features.

[1] Department of Chemistry, Indian Institute of Science Education and Research (IISER), Pune, MH, India. ✉email: s.hotha@iiserpune.ac.in

Sulfated linear polysaccharides consisting of alternating disaccharide units of α-1,4-linked glucosamine and either glucuronic acid or iduronic acids such as heparin (H) and heparin sulfate (HS) are present on the surface of most animal cells, membranes, and extracellular matrices[1,2]. They play a pivotal role in diverse biological pathways including tumor metastasis, cell growth, cell adhesion, wound healing, inflammation, diseases of the central nervous system, etc.[3,4]. Both H and HS are heavily O- and N-sulfated, they belong to glycosaminoglycan polysaccharides and are extracted and isolated from natural animal sources (porcine intestine or bovine lung or sometimes from turkeys, mice, camel, whales, lobsters, etc.)[5]. H and HS are routinely used as anticoagulant drugs during major surgeries such as cardiopulmonary bypass, knee replacement, hip replacement in order to prevent the occurrence of venous thrombosis[6].

Porcine or bovine-derived Heparin has been used in clinics for many decades as an anticoagulant drug due to its strong affinity binding with antithrombin III thereby preventing venous thrombosis[7]. Unfractionated heparin (UFH) (MW$_{avg}$ ~15,000, ~45 monosaccharide chains) and low molecular weight heparin (LMWH) (MW$_{avg}$ ~6000) are marketed for a longtime[8,9]. Several LMWHs are marketed under different trade names such as Enoxaparin, Nadroparin, Reviparin, Dalteparin, Tinzaparin, Certoparin, and Danaparoid depending on the type of depolymerization[10,11]. For example, Enoxaparin is isolated from UFH after β-eliminative cleavage employing alkali through peeling off reaction whereas Nadroparin is obtained by deaminative cleavage employing nitrous acid[12]. Mechanistic investigations revealed that the H binds to antithrombin with high affinity, brings in a conformational change thereby converting it to a rapid (1000×) inhibitor of thrombin (FIIa)[13]. Apart from thrombin, antithrombin interacts with coagulation factor Xa (FXa). LMWHs derived by chemical and/or enzymatic depolymerization procedures from UFH vary in both their relative abilities to enhance the inhibition of FXa and FIIa (anti-FIIa) and in their physicochemical properties. It has been noticed that specific FXa inhibitory activity increases as the mean molecular weight decreases. For example, UFH (MW$_{avg}$ ~15,000) has an anti-FXa/Anti-FIIa activity ratio of 1.0 whereas the same ratio for Enoxaparin (MW$_{avg}$ ~4200) is 3.9 and Bemiparin (MW$_{avg}$ ~3600) was 8.0[12]. However, chemically and enzymatically extracted H and HS from animal sources suffer from microheterogeneity, presence of viral or prion contaminants; and hence, strongly influence their purity and quality from batch to batch[14,15].

The problem manifested into a pinnacle due to the worldwide distribution of contaminated animal-sourced heparin about a decade ago.

Much before, in the 1980s, a unique pentasaccharide domain of heparin was found to be clinically effective as a specific FXa inhibitor[16–18]. This important discovery paved the way for the chemically synthesized pentasaccharide that later led to the launch of the first synthetic anticoagulant antithrombotic Fondaparinux (Arixtra®) 1 in 2004 (Fig. 1)[19]. Fondaparinux (MW = 1725) has well-controlled pharmacokinetic and pharmacodynamics properties, is free from any viral or prion impurities, and importantly, is a specific FXa inhibitor. Extensive structure–property relationship studies proved that essential sulfate and carboxylic acid groups shall be located at opposite sides of the pentasaccharide. Despite its predictable anticoagulant dose and long half-life, Fondaparinux (1) is very expensive compared to H and HS derived from animals as its synthesis demands a long and tedious procedure diminishing the overall efficiency[20]. Indeed, the synthesis of fondaparinux pentasaccharides and other heparin oligosaccharides is a herculean task due to the intricacies involved in the installation and unblocking of multiple orthogonal protecting groups. Since the first synthesis of the pentasaccharide by Petitou in 1987[21], several synthetic strategies have been reported for the heparin fragments involving stepwise glycosylation invoking many protecting groups and glycosylation protocols.

Given the complexity of Fondaparinux, very few total syntheses[22–29] are reported to date and they mainly focused on the following aspects in order to improve the overall yield: (i) optimizing chemistry of individual monosaccharides; (ii) identification of right pairs of orthogonal protecting groups; (iii) stereoselective glycosylation chemistry. On the contrary, HS polymerase, sulfotransferases, and epimerases were employed for the enzymatic synthesis of the Fondaparinux. In spite of these, access to differentially substituted derivatives is very significant for structure-property relationships of the Fondaparinux. Reported synthesis of the pentasaccharide 1 till date have modeled their convergent or linear or multiple one-pot strategies either by 3 + 2 or 3 + 1 + 1 combination of modular saccharide building blocks. A rapid and facile synthetic strategy for compound 1 that also enables the creation of diverse molecular entities that differ in sulfation pattern is still in demand. Accordingly, an efficient synthetic strategy has been envisioned for the synthesis of Fondaparinux pentasaccharide standing on the silver-assisted gold-catalyzed activation of glycosyl carbonates.

## Results and discussion

We targeted the synthesis of Fondaparinux pentasaccharide by employing our recently discovered silver-assisted gold-catalyzed glycosidations[30] on strategically designed monosaccharide building blocks. The synthesis of Fondaparinux pentasaccharide 1 was envisioned from the regioselectively protected pentasaccharide 2. Trisaccharides 3 was chosen as the key precursor for effecting either 3 + 2 glycosylation using disaccharide 4 or 3 + 1 + 1 elongation by coupling iduronate 5 followed by the azido-derivative 6. Trisaccharides 3 can be synthesized by stepwise glycosylation using alkynyl glycosyl carbonates and Au- and Ag-salts as catalysts. Iduronates (5, 10) and glucuronate 8 are envisaged from D-Glucose whereas building blocks 6, 7, and 9 are imagined from D-Glucosamine (Fig. 1). Importantly, building blocks are designed in such a way that the scale-up and convergent synthesis from common precursors shall be accomplished in minimal steps.

**Synthesis of GH disaccharide.** Accordingly, our synthesis endeavor commenced with the identification of a scalable method for the iduronate 5. Chelation assisted nucleophilic addition on a C-5 aldehyde 12 that can be easily accessed from diacetoneglucofuranose derivative 11 was envisioned for the synthesis of idose derivative 13 (Fig. 2A)[31]. The nucleophile shall be chosen in such a way that it can be converted into a carboxylate at a later stage. Hence, as a model reaction, commercially available diacetoneglucofuranose was transformed into aldehyde 14a[32] and was treated with commercially available PhMgBr to notice the formation of phenyl carbinols that are ido-configured 15a and gluco-configured 16a in a ratio of 10:1 with an overall yield of 85% which could be easily separated by silica gel column chromatography. Furanose to pyranose conversion of compound 15a was easily accomplished under acidic conditions to afford pyranosides which were treated with Ac$_2$O and pyridine to get ido-derivative 17a in 74% yield. Oxidation of idopyanoside 17a to iduronate 18a was accomplished with the help of RuO$_4$ which was generated in situ[33,34] from RuCl$_3$ and NaIO$_4$ followed by the esterification using K$_2$CO$_3$ and CH$_3$I (Fig. 2B).

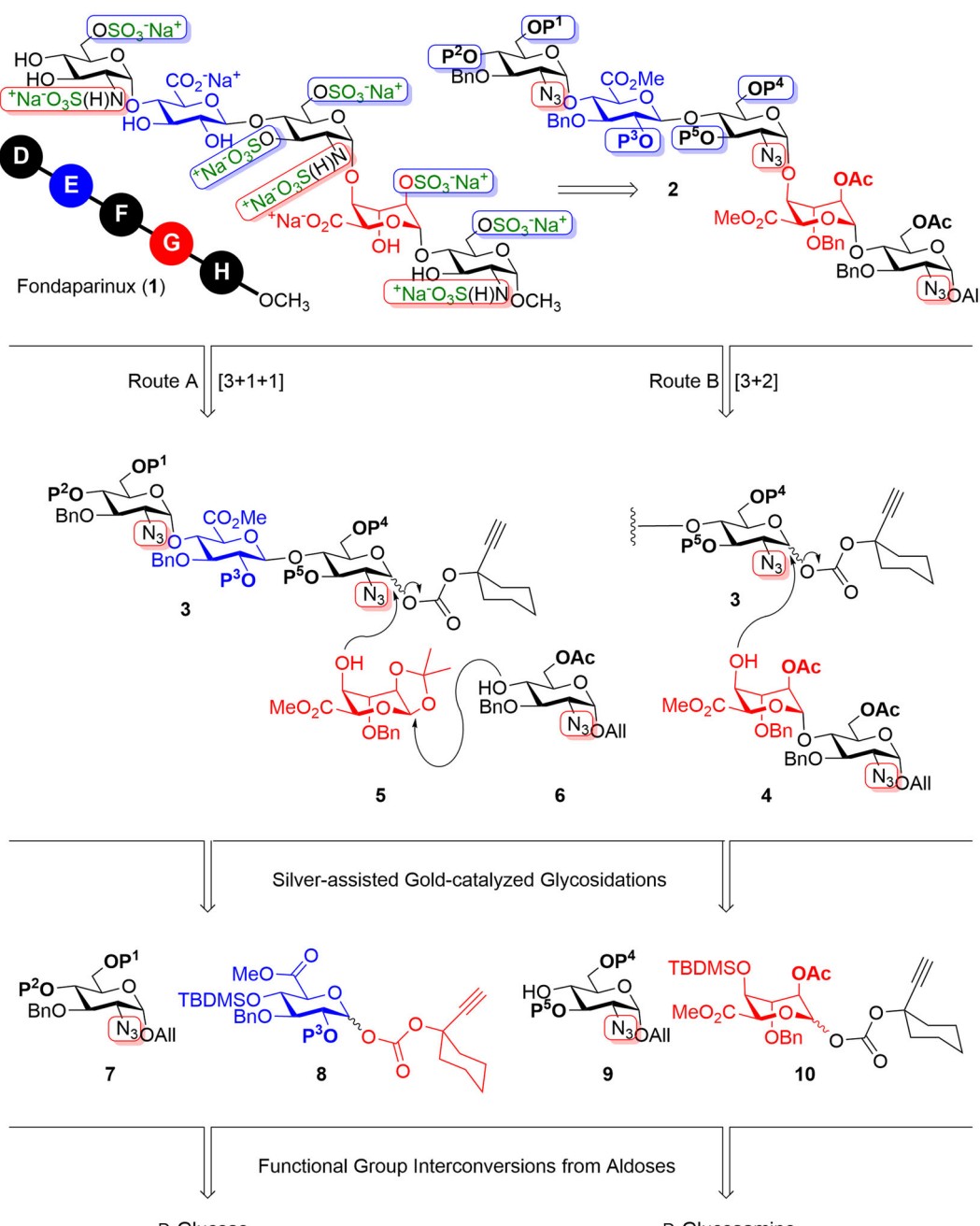

**Fig. 1 Retrosynthesis of the pentasaccharide.** Fondaparinux (**1**) can be synthesized from the advanced intermediate **2** which in turn can be synthesized by either [3 + 1 + 1] using **3**, **5**, **6**, or [3 + 2] manner from compound **4**. Compounds **3**, **4** can be obtained from monosaccharides **7–10**.

Synthesis of target molecule warrants installation of benzyl ether at the C-3 position. Thus, freshly prepared aldehyde **14b** was subjected to Grignard reaction using PhMgBr to obtain 12:1 ratio of ido- (**15b**) and gluco- (**16b**) derivatives in 83% yield. Furanoside **15b** was converted to pyranoside **17b** by the treatment of aq. 90% TFA followed by acetylation under Ac₂O/py conditions. At this stage, oxidation of phenyl group of idoside **17b** in the presence of benzyl ether posed a serious challenge. Several oxidation conditions are tried and failed to affect the regioselective oxidation of the phenyl group and hence could not isolate the iduronate **18b** (Fig. 2B).

Idose synthesis via the chelation assisted Grignard addition strategy is highly beneficial as the synthesis on large scales is feasible[35–38]. Therefore, the addition of 2-Thienylmagnesium bromide[39,40] caught our attention as the thiophene can undergo

oxidation under mild conditions without disturbing the benzyloxy group at the C-3 position. Consequently, freshly prepared 2-thienylmagnesium bromide was treated with aldehyde **14b** to obtain the ido-derivative **19** predominantly in 85% yield (see Supplementary Fig. S13a–c). Acetylation under Ac₂O to obtain acetate **20** followed by the oxidation of the thiophene moiety to the desired acid and subsequent esterification underwent smoothly to afford iduronate in 74% yield over two steps followed by the Zemplén deacetylation afforded compound **21** in 94% yield (Fig. 2C). Furanose to pyranose conversion of iduronate **21** under acidic conditions followed by its conversion to the isopropylidene derivative **5** occurred uneventfully and the remaining C-4 axial hydroxyl group was protected as silyl ether using TBDMSOTf/2,6-lutidine to afford compound **23** that was converted to the much-desired glycosyl donor **10** in two steps.

**Fig. 2 Strategy for the synthesis of iduronic acid (A), synthesis of iduronic acid (B), and synthesis of desired iduronic acid building block (C).** Reagents. **a** PhMgBr, THF, RT, 3 h, 85% (**15a**: **16a** = 10:1), 83% (**15b**:**16b** = 12:1); **b** (i) TFA-H$_2$O (9:1), 25 °C, 15 min, (ii) Ac$_2$O, pyridine, DMAP(cat), 2 h, **17a**: 74%, **17b**: 69%; over two steps; **c** RuCl$_3$·3H$_2$O (5 mol%), NaIO$_4$ (8 eq.), CCl$_4$–CH$_3$CN-H$_2$O (1:1:1.5), then K$_2$CO$_3$, CH$_3$I, DMF, 25 °C, 8 h, **18a**: 72%, **18b**: 0% over two steps. **d** 2-Thienylmagnesium bromide, Et$_2$O, 25 °C, 3 h, 89% (**19a**:**19b**: 10:1); **e** Ac$_2$O, DMAP, 1 h, 25 °C, 95%; **f** (i) RuCl$_3$.3H$_2$O (5 mol%), NaIO$_4$ (8 eq.), hexane-ethylacetate-H$_2$O (1:3:4), then K$_2$CO$_3$, CH$_3$I, DMF, 25 °C, 8 h, 74%; over two steps; (ii) NaOMe, MeOH, 25 °C, 30 min, 94%; **g** TFA-H$_2$O (9:1), 25 °C, 15 min, followed by (**h**) 2-methoxypropene, THF, PTSA, 0-25 °C, 5 h, 55%; over two steps; **i** TBDMSOTf, 2,6-lutidine, CH$_2$Cl$_2$, 25 °C, 15 min, 93%; **j** (i) 75% Dichloroacetic acid (aq.), 0-25 °C, 1 h, 81%; (ii) 1-ethynylcyclohexyl (4-nitrophenyl) carbonate (**24**), DMAP, CH$_2$Cl$_2$, 3 h, then Ac$_2$O, 2 h, 25 °C, 97%; **k** (i) **6**, 8 mol% each of (2,4-($^t$Bu)$_2$C$_6$H$_3$O)$_3$PAuCl (**25**) and AgOTf, CH$_2$Cl$_2$, 4 Å MS, 25 °C, 15 min, 89%, (ii) HF•py, pyridine, 5 h, 0-25 °C, 86%.

**Fig. 3 Synthesis of glucosamine building blocks reagents. a** (i) AcCl, AllOH, 0–80 °C, 16 h, 94%; (ii) PhCH(OMe)$_2$, CSA, DMF, 25 °C, 4 h, 74%; **b** Zn, AcOH:CH$_3$OH:CH$_2$Cl$_2$ (1:2:1), 0–25 °C, 1 h, 77%; **c** azidosulfonylimidazolide, K$_2$CO$_3$, CuSO$_4$•5H$_2$O, MeOH-THF (4:1), 25 °C, 2 h, 82%; **d** (i) NaH, PMB-Cl/BnBr/NapBr, TBAI (cat.), THF, 25 °C, 3 h, **30a**: 81%; **30b**: 87%; **30c**: 84%; or (ii) TBDPSCl, Et$_3$N, DMAP, CH$_2$Cl$_2$, 25 °C, 15 h, **30d**: 78%; or (iii) Ac$_2$O, DMAP, CH$_2$Cl$_2$, 30 min, RT; **30e**: 84%; **e** PTSA, MeOH-CH$_2$Cl$_2$ (1:1), 25 °C, 1 h; then **f** Bz$_2$O/Ac$_2$O, Et$_3$N, CH$_2$Cl$_2$, 12 h; **31a**: 74%, **6**: 75%; over two steps; or **g** TBDPSCl, Et$_3$N, DMAP, CH$_2$Cl$_2$, 25 °C, 15 h; **31b**: 87%, **31c**: 71%, **31d**: 79%, **31e**: 76%, **31f**: 74%; over two steps **h** NaH, NapBr/PMB-Cl/BnBr, TBAI (cat.), THF, 25 °C, 3 h **32a**: 92%, **32b**: 87%, **32c**: 93%; **i** HF•py, pyridine-THF (8:2), 5 h, 0–25 °C, 89%. **j** (i) Ac$_2$O, DMAP, CH$_2$Cl$_2$, 25 °C, 30 min; **32d**: 86%; (ii) LevOH, DIC, DMAP, CH2Cl2, 0–25 °C, 2 h **32e**: 93%; (**k**) (i) PdCl$_2$, MeOH-CH$_2$Cl$_2$ (3:1), 25 °C, 4–8 h; then (ii) **24**, DMAP, CH$_2$Cl$_2$, 25 °C, 3 h **33a**: 79%; **33b**: 68%; **33c**: 78%; **33d**: 69%; **33e**: 71%; over two steps.

Firstly, careful hydrolysis of the isopropylidene group afforded a hemiacetal (see Scheme S1) that was directly treated with 1-ethynylcyclohexyl (4-nitrophenyl) carbonate (**24**)[30] in the presence of DMAP to obtain an anomeric mixture of carbonates, and finally, the C-2-OH was protected as its acetate to obtain donor **10** in 97% yield (Fig. 2C, see Supplementary Scheme S1).

**Synthesis of glucosamine building blocks**. In parallel, synthesis of regioselectively protected glucosamine building blocks commenced with the preparation of Troc-protected glucosamine derivative **26** adopting known procedures[41]. Tetraol **26** was first converted to its allyl α-D-glycoside which was converted to benzylidene **27** by using benzylidenedimethylacetal and the Troc-protecting group was unmasked by Zn-mediated reaction to afford the amino alcohol **28** in 77% yield. An azide at the C-2 position strongly influences the stereochemical outcome of glycosidation in favor of desired 1,2-*cis* or α-glucoside. Therefore, conversion of amine to azide was easily accomplished by the use of freshly prepared azidosulfonylimidazole[42] in CH$_3$OH-THF to afford compound **29** in 82%. At this point, a series of protections on the lone hydroxyl group were considered in order to explore orthogonal protections/deprotections at a subsequent stage. Accordingly, the lone hydroxyl group was protected as *p*-methoxybenzyl ether (**30a**) or benzyl ether (**30b**) or naphthyl ether (**30c**) or silyl ether (**30d**) or acetate (**30e**) using appropriate reaction conditions (Fig. 3). In continuation, benzylidene acetal of

compounds **30a–30e** was hydrolyzed under acidic conditions, regioselective protection of the C$_6$-OH was accomplished to get desired aglycons **31a–31f**, **6** in good yields (see Supplementary Scheme S2).

The C$_6$-silyl ethers were accomplished using silyl chloride and trimethylamine whereas C$_6$-esters were realized by the treatment of diol with corresponding anhydrides at 25 °C (Fig. 3). It has been foreseen that the synthesis of trisaccharide **3** can be achieved by either [2 + 1] or [1 + 2] fashion for which another set of building blocks are required. Therefore, compound **31b** was protected as either naphthyl ether (**32a**), PMB ether (**32b**), or benzyl ether (**32c**) under suitable conditions. Cleavage of the silyl ether of naphthyl ether **32a** followed by the C$_6$-protection afforded other building blocks **32d** and **32e**. Compounds **32a–32e** are quite interesting as they possess orthogonal protecting groups that can be unmasked when desired without affecting the other. In the next sequence, Pd-catalyzed cleavage[43,44] of the allyl glycoside afforded hemiacetals that were conveniently converted to the much-desired carbonate donors **33a–33e** in excellent yields (Fig. 3 and Supplementary Scheme S3).

**Synthesis of glucuronic acid building blocks**. Thus synthesized acetate building block **6** and the building block **10** prepared vide supra were subjected to the silver-assisted gold-catalyzed glycosidation using (2,4-($^t$Bu)$_2$C$_6$H$_3$O)$_3$PAuCl (**25**)/AgOTf to afford a 1,2-*trans* disaccharide in 89% that upon hydrolysis of the silyl

**Fig. 4 Synthesis of glucuronic building blocks. reagents. a** (i) TFA-H$_2$O (9:1), 25 °C, 15 min, (ii) AcCl, AllOH, 0-80 °C, 16 h, 76% over two steps, (iii) PhCH(OMe)$_2$, CSA, DMF, 25 °C, 4 h, 75%, (iv) LevOH, DIC, DMAP, CH$_2$Cl$_2$, 0–25 °C, 2 h, **35a**: 92% or Ac$_2$O/ BzCl, Pyridine, DMAP (cat), 0–25 °C, 2 h, **35b**: 95%, **35c**: 92%; **b** (i) PTSA, MeOH-CH$_2$Cl$_2$ (1:1), 25 °C; then (ii) TEMPO, BAIB, CH$_2$Cl$_2$-H$_2$O (2:1), 25 °C, 3 h followed by K$_2$CO$_3$, MeI, DMF, 25 °C, 8 h, **36a**: 66%, **36b**: 66%, **36c**: 67% over three steps; **c** (i) Ac$_2$O, DMAP, CH$_2$Cl$_2$, 25 °C, 30 min, **37a**: 87%; (ii) LevOH, DIC, DMAP, CH$_2$Cl$_2$, 0–25 °C, 2 h, **37b**: 86%; (iii) TBDMSOTf, 2,6-lutidine, CH$_2$Cl$_2$, 25 °C, 20 min, **37c**: 93%; **d** (i) PdCl$_2$, MeOH–CH$_2$Cl$_2$ (3:1), 25 °C, 4–8 h; (ii) **24**, DMAP, CH$_2$Cl$_2$, 25 °C, 3 h, **38a**: 72%, **38b**: 72%, **38c**: 71% over two steps.

ether under HF•py conditions gave the key disaccharide **4** (Fig. 2C). Next in our journey is the synthesis of glucuronate donor **8**. Here again, a diversified set of glucuronyl donors (**38a**–**38c**) were envisioned in order to perform reciprocal donor–acceptor studies with orthogonal protecting groups (Fig. 4).

The path started with the commercially available benzyl protected diacetone glucofuranose **34** which was converted to the pyranoside **35a**–**35c** in three steps. Stirring of compound **34** in aqueous TFA at room temperature facilitated hydrolysis of isopropylidene and subsequent treatment with allyl alcohol in acidic conditions 0-80 °C afforded allyl α-D-glucopyranoside whose C4 and C6 alcohols were locked as benzylidene. Further, the remaining C-2-OH was protected as either levulinoate (**35a**) or acetate (**35b**), or benzoate (**35c**) under standard conditions. The presence of ester at C-2 would assist in neighboring group participation to obtain 1,2-*trans* glucuronide.

In continuation, the benzylidene of allyl pyranosides **35a**–**35c** was hydrolyzed using PTSA/MeOH–CH$_2$Cl$_2$ to obtain a diol and the resulting primary hydroxyl moiety was oxidized under TEMPO/BAIB conditions[24,25] to obtain an acid that was protected as their methyl esters **36a**–**36c** (see Scheme S3). The lone hydroxyl group of ester **36a**–**36c** was orthogonally protected to obtain esters **37a**–**37c**. Furthermore, allyl glucuronides **37a**–**37c** were hydrolyzed under PdCl$_2$/MeOH–CH$_2$Cl$_2$ conditions and subsequently treated with the carbonate reagent **24** to get donors **38a**–**38c** in excellent yields (Fig. 4). Synthesis of the trisaccharide **3** is contingent on developing protocols and synthesizing all the identified building blocks in enough quantities. Trisaccharide **3** synthesis commenced with optimization of conditions for the stereoselective synthesis of D–E disaccharide using donors **33a**–**33e** and aglycons **36a**–**36c** prepared to vide supra. All glycosidations between azidoglucosyl donors and acceptor were conducted employing Au-phosphite (**25**) and AgOTf in CH$_2$Cl$_2$ containing 4 Å MS powder to afford the disaccharide **39a**–**39i** (Table 1).

*Synthesis of DEF trisaccharide.* Our explorations on reciprocal donor–acceptor selectivity studies commenced with the silver-

assisted gold-catalyzed glycosidation with the donor **33a** and acceptor **36a** that have orthogonal protecting groups TBDPS- and Nap- on glycosyl donor and Lev-, Benzyl-moieties to afford D-E disaccharide **39a** in 77% yield with 4:1 (α:β) ratio (Entry 1). Changing the protecting group to the acetate as in **36b** did not alter the α:β ratio of disaccharides **39b** (Entry 2), Next, the glycosylation between donors **33b**–**33e** and acceptor **36b** showed marginal improvement in the α:β ratio with little difference in the yield of the resulting disaccharides **39c**–**39f** (Entries 3–6). However, a quantum jump in the α:β ratio to 8:1 was noticed when the glycosylation was carried out between the glycosyl donor **33a** and **33b** at −20 °C with a very high yield as well (Entry 7). Though the selectivity got improved, the 1,2-*cis* or α- linked disaccharide and the unwanted β-isomer are noticed to have similar polarities and hence, demanded multiple column chromatographic separations thereby compromising the overall process efficiency.

Therefore, the glycosylation between the donor **33a**–**33c** and acceptor **36c** was performed at 25 °C to obtain the disaccharide **39g**–**39i** with a similar α:β ratio of D–E disaccharides (Entry 8–10). To our satisfaction, a good Rf difference was noticed in compounds **39g** and **39i** thereby facilitating the easy purification. Further, the temperature dependence of the glycosylation between donor **33a**, **33c**, and acceptor **36c** at −20 and −40 °C was performed to notice that the best stereoselectivity and yield were obtained with the donor possessing C6-OTBDPS and C4-OBn ethers (**33c**) and acceptor having C2-OBz (Entries 11–14). Thus the reciprocal donor-acceptor studies showed that glycosylation shall be carried out at −40 °C using glycosyl donor **33c** and acceptor **36c** (Entry 14, Table 1). The next milestone is the synthesis of D-E-F trisaccharide. In this direction, allyl moiety of the disaccharide **39i** was smoothly cleaved off by Pd-catalyzed conditions to afford hemiacetals (see Supplementary Scheme S4) which were treated with the carbonate reagent **24** in the presence of DMAP in CH$_2$Cl$_2$ to afford the desired D-E disaccharide donor **40** that can be used for the synthesis of D-E-F trisaccharide (Fig. 5).

Next, attempts to synthesize trisaccharide from the disaccharide **40** and **31c**–**31f** failed to give the desired trisaccharide; instead, resulted in the isolation of 1,2-eliminated compound **41**

**Table 1 Stereoselective synthesis of D–E disaccharide.**

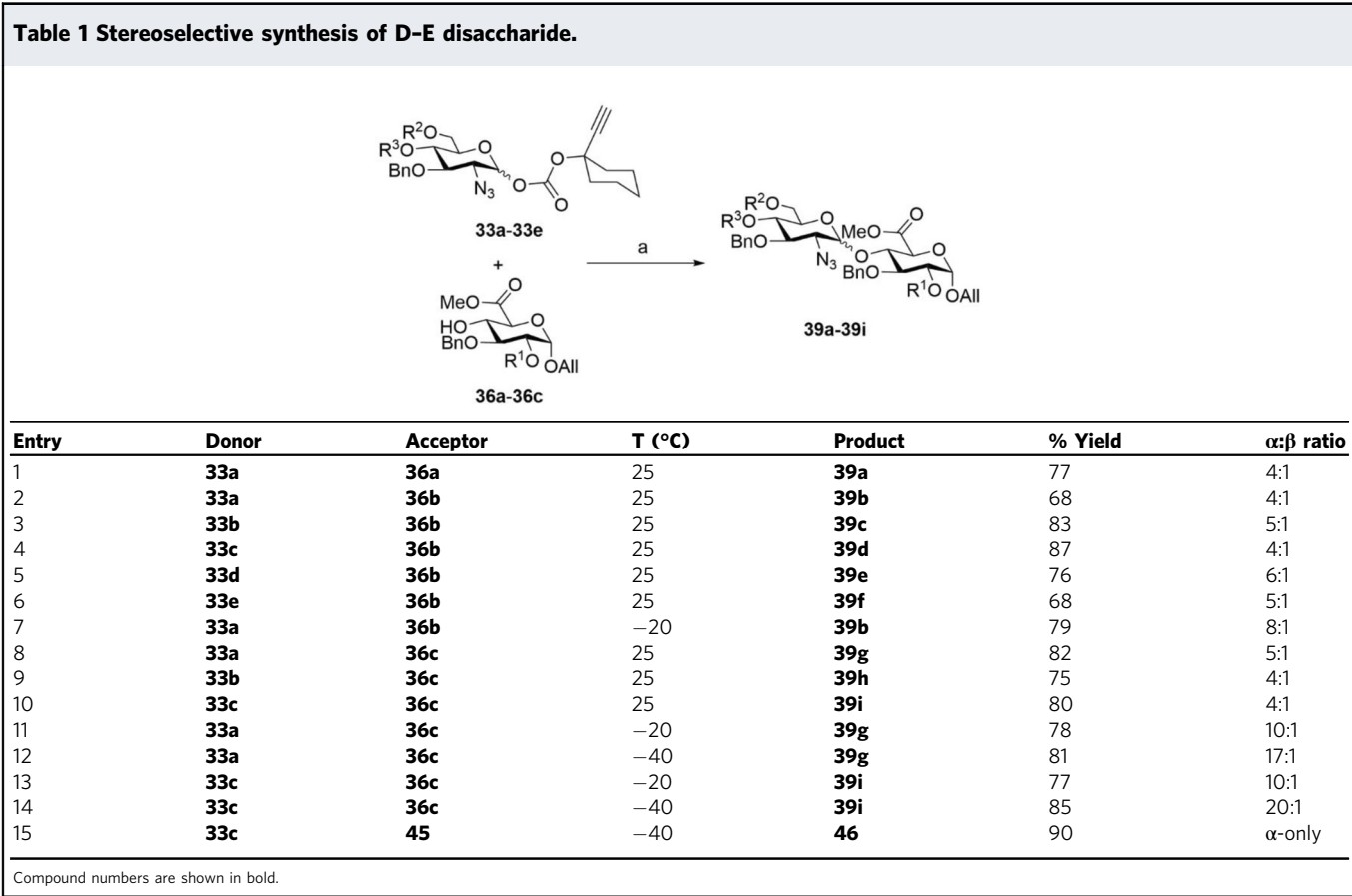

| Entry | Donor | Acceptor | T (°C) | Product | % Yield | α:β ratio |
|---|---|---|---|---|---|---|
| 1 | 33a | 36a | 25 | 39a | 77 | 4:1 |
| 2 | 33a | 36b | 25 | 39b | 68 | 4:1 |
| 3 | 33b | 36b | 25 | 39c | 83 | 5:1 |
| 4 | 33c | 36b | 25 | 39d | 87 | 4:1 |
| 5 | 33d | 36b | 25 | 39e | 76 | 6:1 |
| 6 | 33e | 36b | 25 | 39f | 68 | 5:1 |
| 7 | 33a | 36b | −20 | 39b | 79 | 8:1 |
| 8 | 33a | 36c | 25 | 39g | 82 | 5:1 |
| 9 | 33b | 36c | 25 | 39h | 75 | 4:1 |
| 10 | 33c | 36c | 25 | 39i | 80 | 4:1 |
| 11 | 33a | 36c | −20 | 39g | 78 | 10:1 |
| 12 | 33a | 36c | −40 | 39g | 81 | 17:1 |
| 13 | 33c | 36c | −20 | 39i | 77 | 10:1 |
| 14 | 33c | 36c | −40 | 39i | 85 | 20:1 |
| 15 | 33c | 45 | −40 | 46 | 90 | α-only |

Compound numbers are shown in bold.

**Fig. 5 Synthesis of D–E disaccharide donor.** Reagents. **a** (i) PdCl$_2$, MeOH–CH$_2$Cl$_2$ (3:1), 25 °C, 4 h, 95%, (ii) **24**, DMAP, CH$_2$Cl$_2$, 25 °C, 3 h, 85%.

as a major product (Entries 1–4, Table 2). Discouraged by these results while executing the DE + F strategy for the DEF trisaccharide prompted us to explore its synthesis by D + EF strategy. Accordingly, **38a** + **31e** glycosylation was performed and noticed the formation of the 1,2-orthoester **42** (Entry 5, Table 2). However, **38b** + **31c** produced the desired EF disaccharide **43a** albeit in poor yield due to the loss while separating the desired EF disaccharide from a myriad of uncharacterized products that formed (Entry 6, Table 2). Although **38c** + **31c** afforded **43b** in 80% but the polarity of the acceptor **31c** and the product **43b** were almost similar thereby warranting multiple flash purifications (Entry 7, Table 2). In addition, hydrolysis of the TBDMS-ether in the presence of TBDPS-ether also diminished the net yield in the subsequent step. Finally, **38c** + **31a** resulted in the desired EF disaccharide **44** with 1,2-*trans* interglycosidic linkage in excellent 91% yield (Entry 8, Table 2).

**Synthesis of Fondaparinux pentasaccharide.** Continuing our synthesis, cleavage of silyl ether of the E–F disaccharide **44** was smoothly achieved under HF•py conditions to obtain alcohol **45** which was subjected to the silver-assisted gold-catalyzed

glycosidation with glycosyl donor **33c** to afford the D-E-F trisaccharide **46** in 90% yield. Gratifyingly, trisaccharide **46** was noticed that glycosidation happened in stereoselective fashion resulting in a single 1,2-*cis* or α-anomer only at −40 °C; an anomeric mixture of trisaccharides was noticed above −20 °C. Trisaccharide **46** was extrapolated to the carbonate donor **47** in two aforementioned steps by Pd-catalyzed hydrolysis of the allyl ether and subsequent treatment with the carbonate reagent **24** (Fig. 6).

Trisaccharide donor **47** was split into portions and the first portion was treated with the iduronate **5** under Au/Ag-catalysis conditions to afford the tetrasaccharide **48** in 91% within 15 min at 25 °C. Gratifyingly, the glycosidation between **47** and **5** resulted in complete α-selectivity at 25 °C presumably due to the presence of the isopropylidene of the aglycone. The remaining portion of the trisaccharide donor **47** was treated with the disaccharide **4** that was prepared vide supra to obtain the desired pentasaccharide **50a** reminiscent of the Fondaparinux in 86% yield (Fig. 6, see Supplementary Fig. 98a–p)[14].

In parallel, hydrolysis of the isopropylidene and PMB-ether present in the tetrasaccharide **48** was effected under acidic conditions and C-1 position was transformed into the alkynyl

**Table 2 Studies on glucuronic acid glycosidations.**

| Entry | Donor | Acceptor | Product |
|-------|-------|----------|---------|
| 1. | **40** | **31c** | **41 (45-70%)** |
| 2. | **40** | **31d** | |
| 3. | **40** | **31e** | |
| 4 | **40** | **31f** | |
| 5. | **38a** | **31e** | **42 (71%)** |
| 6. | **38b** | **31c** | **43a (35%)** |
| 7. | **38c** | **31c** | **43b (80%)** |
| 8. | **38c** | **31a** | **44 (91%)** |

Compound numbers are shown in bold.

carbonate and the remaining diol was protected as diacetate to afford compound **49**. Finally, donor **49** was coupled with the acceptor **6** affording the desired pentasaccharide **50b** in 75% yield (see Supplementary Fig. 101a–p).

Pentasaccharide **50a** was split into two portions and subjected to selective deprotections. Unmasking of the *p*-methoxybenzyl group of the F-unit was successfully accomplished by the action of DDQ in $CH_2Cl_2$–$H_2O$ at 25 °C in 30 min to afford compound **51** in 85% yield. The monosulfated derivative of compound **51**, which is otherwise very difficult to synthesize, shows a very significant improvement in the anticoagulant activity. Further, the fluoride ion mediated cleaving of the silyl ether afforded the diol **53** in 81% yield. In parallel, the second portion was subjected to the deprotection of silyl ether first to afford pentasaccharide **52** followed by the deprotection of the PMB-ether afforded the compound **53** (see Supplementary Fig. 104a–p). These regioisomeric hydroxyls will be highly useful for the sulfation to study their biological properties.

## Conclusions
In summary, a flexible, modular, and highly efficient synthetic strategy has been developed for the synthesis of Fondaparinux pentasaccharide that stands on the gold–silver catalyzed glycosidation chemistry. This unique route is constructed upon the coupling of thoughtfully identified monosaccharide building blocks which can be synthesized from shared precursors. A scalable chelation assisted method for the synthesis of protected iduronic acid was accomplished using thiophene as a surrogate for the carboxylic acid. All glycosidations are conducted under silver-assisted gold-catalyzed conditions utilizing recently discovered alkynylcyclohexyl carbonate donor chemistry. All glycosidations were optimized to give exclusive stereoselectivity thereby minimizing the tedious purifications. This strategy offers a new catalytic route to the synthesis of mono 3-*O*-sulfation at F-ring of fondaparinux pentasaccharide which is hitherto very difficult to obtain[45]. This significantly improved route for fondaparinux pentasaccharide illustrates a new set of building blocks, the utility of gold-catalyzed glycosidations, and promised to yield orthogonally protected hydroxyl groups that facilitate installation of sulfates in a regiodefined fashion. The 3 + 2 route is preferable for the synthesis of fondaparinux pentasaccharide; however, 3 + 1 + 1 is suitable for diversification of the core structure for structure–function studies. Further research on the exploitation of this strategy for the synthesis of other glycosaminoglycan derivatives of therapeutic significance is currently underway.

## Methods
**General methods**. Unless otherwise noted, materials were obtained from commercial suppliers and were used without further purification. All metal salts were purchased from Sigma-Aldrich. Unless otherwise reported all reactions were performed under Argon atmosphere. Removal of the solvent in vacuo refers to the distillation using a rotary evaporator attached to an efficient vacuum pump. Products obtained as solids or syrups were dried under a high vacuum. Analytical thin-layer chromatography was performed on pre-coated silica plates ($F_{254}$, 0.25 mm thickness); compounds were visualized by UV light or by staining with anisaldehyde spray. Optical rotation was measured on a digital polarimeter. IR spectra were recorded on a Fourier-transform infrared spectrometer. Nuclear magnetic resonance (NMR) spectra were recorded either on a 400 or a 500 or a 600 MHz with $CDCl_3$ or $CD_3OD$ as the solvent and TMS as the internal standard. High-resolution mass spectroscopy was performed using an electrospray ionization time-of-flight mass analyzer. Low-resolution mass spectroscopy was performed on ultra-performance liquid chromatography–mass spectrometry with SWADESI-TLC interface. For NMR analysis and high-resolution mass spectrometry of the compounds in this article, see Supplementary Figs. 1–104. Details experimental methods including all intermediate structures (see Supplementary Schemes S1–S4) are provided in the Supplementary Information.

### Experimental methods
*Selective oxidation of aryl/heteroaryl moieties*. To a solution of **17a** (1 mmol) in $CCl_4$:$CH_3CN$ (30 mL, 1:1) {for compound **20**—*n*-hexane:ethyl acetate 30 mL, 1:3} for was added solution of $NaIO_4$ (8 mmol) in $H_2O$ (30 mL) at 25 °C and stirred for 10 min. $RuCl_3•3H_2O$ (5 mol%) was added and the mixture was stirred vigorously for 24 h. NaCl was added to saturate the aqueous layer and extracted with $CH_2Cl_2$. Combined organic phases were washed with brine solution, dried over $Na_2SO_4$, and concentrated in vacuo. The resulting crude residue was redissolved in anhydrous DMF (15 mL) was added 1.5 equivalent of $K_2CO_3$. After stirring for 15 min, iodomethane (2 mmol) was added under an argon atmosphere and the mixture was stirred at 25 °C for 8 h in a dark place. After complete consumption, the reaction was quenched by the addition of an excess amount of saturated solution of $Na_2SO_3$, followed by the addition of water and extracted with EtOAc. The combined organic phases were washed with brine, dried over anhydrous $Na_2SO_4$, and concentrated under reduced pressure. The crude residue was purified by silica gel column chromatography. This method was utilized for the preparation of compounds **18a** and **21**.

*Selective oxidation of 1° alcohol to esters*. To a biphasic solution of the 4,6-diol (1 mmol) in 2:1 $CH_2Cl_2$–water (10 mL) was added (diacetoxyiodo)benzene (2.5 mmol) and TEMPO (0.2 mmol) simultaneously, stirred vigorously at 25 °C. After 3 h, the reaction was quenched by the addition of a saturated aqueous solution of $Na_2SO_3$ and extracted with $CH_2Cl_2$, combined organic phases were washed with brine (10 mL), dried over $Na_2SO_4$, and concentrated in vacuo. The crude product was redissolved in anhydrous DMF (5 mL) and added 1.5 equivalents of $K_2CO_3$. After stirring for 15 min, iodomethane (2 eq) was added dropwise under an argon atmosphere and stirred for 8 h at 25 °C in a dark place. After complete consumption, the reaction was arrested by adding a saturated aqueous solution of $Na_2SO_3$. Extracted with EtOAc, combined organic layers were washed with brine, dried over anhydrous $Na_2SO_4$, concentrated under reduced pressure. The crude residue was purified by silica gel column chromatography using ethyl acetate and n-hexane as mobile phase to obtain the desired product. This method was utilized for the preparation of compounds **36a–36c**.

*Grignard reaction*. Aldehyde **14a**, **14b** (1 mmol) was dissolved in 5 mL of the anhydrous THF and added slowly to the freshly prepared Grignard reagent (1.5 mmol) at 0 °C under argon atmosphere. The reaction mixture was allowed to stir at 25 °C for 3 h. After completion, the reaction was quenched by the addition of

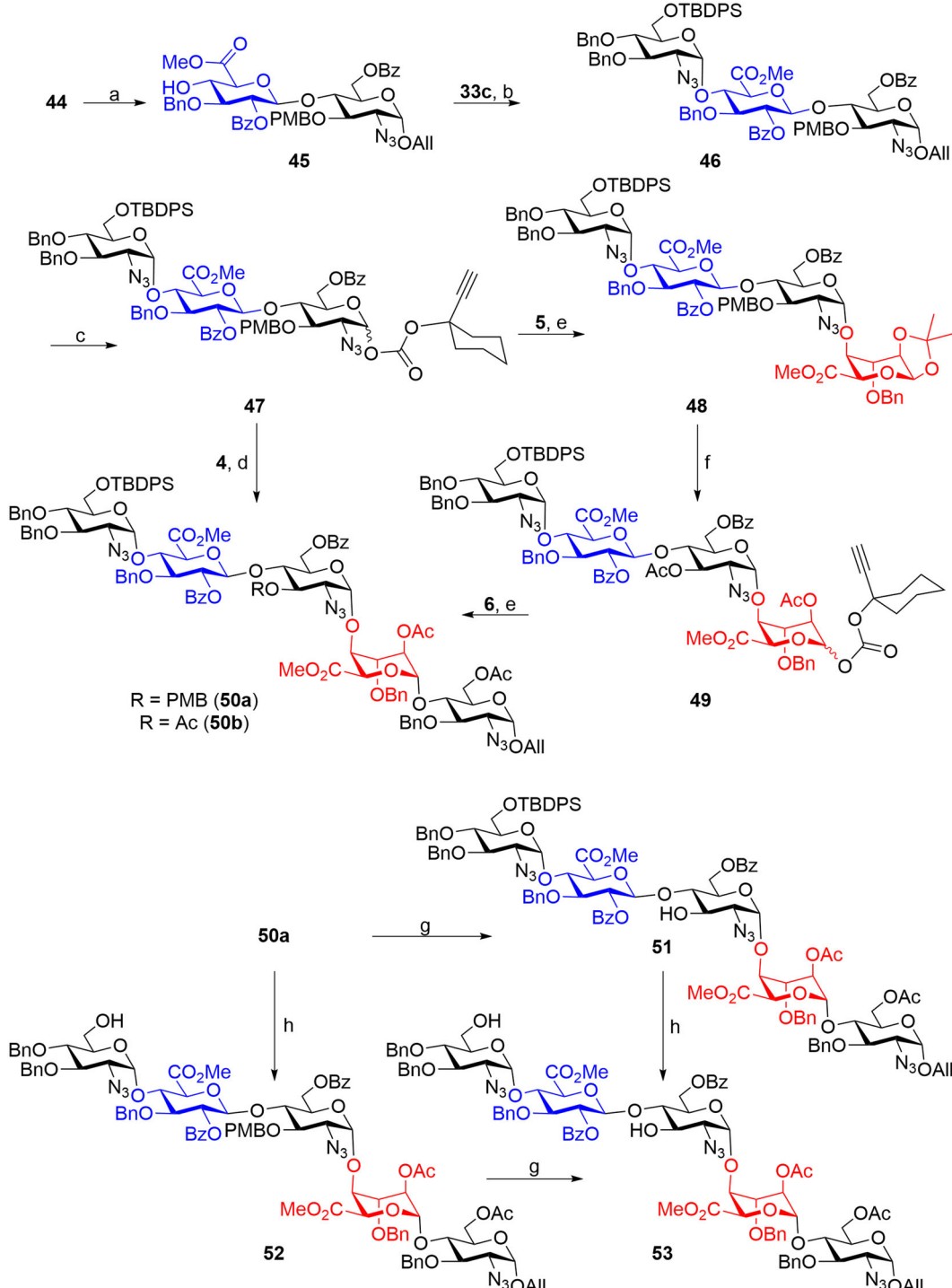

**Fig. 6 Synthesis of fondaparinux pentasaccharide reagents. a** HF•py, pyridine, 0–25 °C, 5 h, 89%; **b** 15 mol% each of **25** and AgOTf, CH₂Cl₂, 4 Å MS, −40 °C, 1 h, 90% (α only); **c** (i) PdCl₂, MeOH–CH₂Cl₂ (3:1), 25 °C, 4 h, 87%; (ii) **24**, DMAP, CH₂Cl₂, 25 °C, 3 h, 88%; **d** 10 mol% each of **25** and AgOTf, CH₂Cl₂, 4 Å MS, −10 °C to 25 °C, 30 min, 86% (α only); **e** 8 mol% each of **25** and AgOTf, CH₂Cl₂, 4 Å MS, 25 °C, 15 min; **48**: 91% (α only), **50b**: 75%; **f** (i) 75% Dichloroacetic acid (aq), 0 °C, 1 h, 68%, (ii) **24**, DMAP, CH₂Cl₂, 25 °C, 3 h, then Ac₂O, 25 °C, 2 h, 71% over two steps. **g** DDQ, CH₂Cl₂: H₂O (10:1), 25 °C, 30 min, **51**: 85%, **53**: 76%; (**h**) HF•py, pyridine, 0–25 °C, 5 h, **52**: 83%, **53**: 81%.

saturated aqueous ammonium chloride solution, water (20 mL) and extracted with EtOAc (3 × 10 mL). The combined organic phases were washed with brine solution (10 mL), dried over anhydrous Na₂SO₄, and concentrated in vacuo. The crude residue was purified by silica gel column chromatography using EtOAc and hexane (30–35%) as a mobile phase to afford aryl carbinols as Ido- or Glc- isomers. Grignard reagent preparation: To a freshly activated Mg-metal (1.5 mmol) and 10 mL of the anhydrous THF in two necks round bottom flask equipped with a condenser was added aryl/heteroaryl bromide (1.0 mmol) slowly under argon

atmosphere and the mixture was stirred at 70 °C for 1 h. This method was utilized for the preparation of compounds **15**, **16**, and **19**.

*Preparation of glycosyl carbonate donors from allyl glycosides. Hemiacetals from allyl glycosides.* To a biphasic solution of the allyl glycoside (1.0 mmol) in 3:1 CH₃OH: CH₂Cl₂ (20 mL) was added 0.15 equivalent of PdCl₂ and the reaction mixture was stirred for 4-8 h at 25 °C, the reaction was quenched by adding an excess of Et₃N and filtered through a bed of Celite®. The filtrate was concentrated in vacuo and the

crude residue was subjected to silica gel column chromatography using ethyl acetate and *n*-hexane as mobile phase to obtain the desired hemiacetal. This method was utilized for the preparation of compounds.

*Synthesis of glycosyl carbonates.* To a solution of glycosyl hemiacetal (1.0 mmol) in anhydrous $CH_2Cl_2$ (5 mL) was added DMAP (1.5 eq) and ethynylcyclohexyl (4-nitrophenyl) carbonate **24** (1.2 eq), the reaction mixture was stirred at 25 °C for 3 h. After complete consumption of hemiacetals, the reaction mixture was concentrated in vacuo and subjected to silica gel column chromatography using EtOAc and hexane as mobile phase.

This two-step procedure is adopted for the synthesis of donors **33a–33e**, **38a–38c**, **40**, and **47**.

*Silver-assisted gold-catalyzed glycosidation.* To a solution of glycosyl donor (1.0 mmol) and acceptor (0.9 mmol) in anhydrous $CH_2Cl_2$ (5 mL) was added freshly activated 4 Å MS powder (0.4 g) at 25 °C under argon atmosphere. After 15 min of vigorous stirring at 25 °C {−40 °C for compound **46**), chloro[tris(2,4-di-t-butyl phenyl)phosphite]gold(I) (**25**) (8 mol%) and AgOTf (8 mol%) were added simultaneously to the reaction mixture and stirred for 15 min. After completion, the reaction mixture was quenched by adding an excess of $Et_3N$ and filtered through a bed of Celite®, the filtrate was concentrated in vacuo and the crude residue was purified by silica gel column chromatography using ethyl acetate and hexane as mobile phase.

This procedure was followed for compounds **S5**, **39a–39i**, **41–44**, **46**, **48**, **50a**, and **50b**.

## Data availability

The authors declare that some of the data supporting the findings of this study are available in its supplementary information files. Detailed experimental methods are provided in the Supplementary Information file. All data is available from the authors upon reasonable request.

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

## Acknowledgements

N.K. and Y.S. acknowledge the research fellowship from CSIR-UGC-NET and G.W. thanks financial support from IISER Pune. S.H. thanks the financial support from DST, New Delhi (EMR/2017/003823). Authors thank DST-FIST funds for high field NMR facility at IISER Pune. The authors thank Mr. Nitin Dalvi, Mr. Sandeep Mishra, and Prof. Jeetender Chugh for high field NMR.

## Author contributions

All experiments were carried out by G.W. with the assistance of N.K., Y.S. under the guidance of S.H. The paper was written through the contributions of all authors. All authors have given approval to the final version of the paper. This research is supported financially by DST-SERB (EMR/2017/3823) for S.H.

## Competing interests

A patent has been filed. The following are the details: Title of the invention: SILVER ASSISTED GOLD CATALYSIS FOR THE PREPARATION OF FONDAPARINUX PENTASACCHARIDE AND INTERMEDIATES, Applicant: Indian Institute of Science Education and Research, Pune; Inventors: Srinivas Hotha, Gulab Walke, Niteshlal Kasdekar, Yogesh Sutar. Application No. 202021050409. Filed on 19.11.2020 in Mumbai. All authors in this paper are part of the patent application and hence, have competing financial interests.
