## [Peer Review File · Communications Chemistry]

Reviewers' comments:

Reviewer #1 (Remarks to the Author):

Comments: reject

In this work, Srinivas Hotha and coworkers have synthesized total protective fondaparinux pentasaccharide analogue backbone based on the Ag/Au catalytic strategy. But it is not a simple, efficient synthetic strategy. There are several designed defects in this work:

1. It is not a complete work because the heavily sulfated molecule, fondaparinux analogue, has not been synthesized.
2. Only a fondaparinux analogue could be obtained from the total protective pentasaccharide. In order to get the beta-isomer of GlcA glycosidation products, a Bz, Ac or other acyl group must be assembled on the 2-OH because of the strategy defects, however, the 2-OH on GlcA was not sulfated. So insightfully, the catalytic strategy is not suitable for the synthesis of fondaparinux.
3. And a Me group was installed on the reducing end rather than an All group on fondaparinux molecule.
4. The Troc of the building block of H was not a suitable protecting group, because of several unnecessary steps were needed. And in some other reported papers, the Cbz was selected as an appropriate protecting group, because it can be removed by H₂ along with Bn and N₃.
5. In order to using the Au/Ag catalytic method, the All must be removed and subsequently the leaving group must be installed on the donor, and several unnecessary steps were needed. So it is not a simple method. Furthermore, the Au and Ag are precious metals, so it is not economical for scalable synthesis of Fonda.

Besides the above five, there are a lot of mistakes in the paper:

1. The title in Supporting Information is different from the title in article.
 2. Line 107, the 18a need to be changed to 18a, line 163, the 30a-30e need to be changed to 30a-30e. and please carefully check the similar mistakes also in this paper.
 3. In the PDF format, almost all the "oC" was changed to " ".
 4. Recently, two papers about the synthesis of Idraparinux and Arixtra have been reported, so these two papers needed to be cited in the article: Journal of the American Chemical Society, 2019, 141, 26, 10309-10314 and Org. Chem. Front., 2019, 6, 3116-3120.
- This work was not recommended published on this journal.

Reviewer #2 (Remarks to the Author):

This manuscript describes the formal synthesis of the anticoagulant Fondaparinux pentasaccharide by employing catalytic promoting system for all glycosidations. The synthesis of heparin and heparan sulfate remains a challenge in carbohydrate chemistry. The low degree of the accessibility to monosaccharides involved in them such as glucuronic acid and iduronic acid and the difficulty of optimizing pairs of orthogonal protecting groups to attain efficient assembly of oligosaccharide sequences in a stereoselective manner has defied many challenges of the chemical synthesis of heparin and heparan sulfate. Therefore, anti coagulant Fondaparinux that is the target molecule in this study has been chemically synthesized by few groups despite the strong demand for its clinical use.

In the present study, the authors have succeeded in highly stereoselective and efficient synthesis of orthogonally protected Fondapaninux pentasaccharides for the first time, which will allow for structure-activity relationship study of Fondapaninux. The authors showcased the efficacy of the silver assisted gold catalyzed glycosidation developed by the author's group, which provided high yields and stereoselectivity in all the glycosidation reactions, almost of which exceed the yields reported in the literatures. The iduronic acid synthesis via the chelation assisted Grignard addition strategy is also interesting. This method will be very useful for large scale preparation. It is overwhelming that 120 mg of oligosaccharide intermediate was synthesized by the authors. I believe this method will fulfill the demand for large scale synthesis of the Fondapaninux and its

analogs. The experimental procedures are detailed and useful for others to reproduce the work. I am happy to recommend this manuscript for publication in Communication Chemistry due to major contribution to this field after revision by considering the following comments.

1. p7, line 145: the conversion into 5 does not seem "uneventfully occurred" due to the middle conversion yield (55%). Additional statement about the byproducts should be involved in the main text.
2. Scheme 2, reaction j: The reason of the use of dichloroacetic acid should be mentioned in the main text.
3. 33e (6-O-Lev-GlcN3) will also fit with the reciprocal strategy like 31a (6-O-Bz-GlcN3), which in addition makes selective deprotection enable in pentasaccharide. Is there any reason why 6-O-Lev-GlcN3 acceptor was not used in the glycosidation examinations in Table 2?
4. p 14, line 261: the statement ' allyl moiety of the disaccharide 39i was smoothly hydrolyzed....' is not correct. This reaction is not hydrolysis.
5. The subscripted numbers with R (R1, R2) in the schemes should be superscripted.

Silver Assisted Gold Catalysis for the Anticoagulant Fondaparinux Pentasaccharide

Gulab Walke, Niteshlal Kasdekar, Yogesh Sutar, Srinivas Hotha

Hotha and co-workers describe their silver-assisted gold catalysis for the anticoagulant fondaparinux pentasaccharide synthesis utilizing a 3 + 1 + 1 and 3 + 2 strategy.

To date, several scientific groups have made the exact same target pentasaccharide molecule, more or less, using their own unique in-house chemistry. The synthetic routes applied to obtain the monosugar units are well-known as the authors have reported these in the bibliographic references (see references 22-27).

The manuscript, itself, is well-written. Two iduronates were prepared, cleverly, and this author has to be given credit for that. Many derivatives were also made for **30, 31, 33, 36, 37, 38** and **39**.

This reviewer, however, understands that the authors did a lot of work on donor/acceptor selectivity studies to get suitable amounts of product after glycosylation. The authors found that **39i** is the best disaccharide to further elaborate their chemistry with. However, it failed to give the desired trisaccharide. After further standardization, the team was then able to show that **44** was the disaccharide to proceed onward with. From **44**, the authors prepared trisaccharide **47** which was split into two parts to ultimately achieve pentasaccharide **50** which was again split into two parts to get partially deprotected pentasaccharide **53**.

The authors conclude that these regioisomeric hydroxyls will be highly useful for the sulfation to study their biological properties. However, this reviewer believes to get final deprotection with suitable sulfonated is very necessary to get high standard publication. All the shown spectral quality is very good but 2D spectra is necessary, once again, for at least from trisaccharide unit. So, the authors are being requested to provide that spectra and as well as show the complete synthesis of Fondaparinux (**1**) as it is shown in Scheme 1.

This reviewer does not recommend publication in Journal of Communications Chemistry based on the manuscripts current state, but with a lengthy time-frame to make the paper more mature, the journal should reconsider reviewing it.

Reviewer Concerns:

- a)** Spectral data needs to be reported with specific numbers. Please mention specific numbers (ppm shifts) for **ALL** ^1H or ^{13}C belonging to each sugar.
- b)** Add this as reference: Page 4 line 75: Preactivation-based, iterative one-pot synthesis of anticoagulant pentasaccharide fondaparinux sodium [*Org. Chem. Front.*, 2019,6, 3116-3120]

- c)** Page 7, line 148: Is that 27 reference correct for this sentence? Or should it be 28?
- d)** Page 3, Scheme 1. In retrosynthetic scheme use '**number 2**' for protected only not for the deprotected pentasaccharide.
- e)** From **17a** to **18a** conversion: Reviewer would like to see the plausible mechanistic explanation. Furthermore, provide a possible reason for not forming product in the case of benzyl protected C-3 position.
- f)** In page 15: line 288-294 and page17, line 304-308: Which route is better to achieve the synthesis of compound **50**? The authors need to conclude that one.

Point-wise Response to Reviewers' Observations

{COMMSCHEM-20-0013-T}

Reviewer 1:

Thank you very much for your comments about our manuscript for publishing in Communications Chemistry.

General: In this work, Srinivas Hotha and coworkers have synthesized total protective fondaparinux pentasaccharide analogue backbone based on the Ag/Au catalytic strategy. But it is not a simple, efficient synthetic strategy.

Answer: We believe that it is simple compared to what is existing out there in the literature for the following reasons: (i) the route stands on the stable glycosyl donor chemistry; (ii) catalytic glycosidations that are high yielding; (iii) the purifications are easier due to the choice of protecting groups and the polarity differences; (iv) all the glycosidations occur in a stereoselective fashion thereby reducing the tedious and lengthy column chromatographies; (v) finally, one of the first catalytic routes for the fondaparinux pentasaccharide. (vi) glycosidations can be performed easily.

Q1) It is not a complete work because the heavily sulfated molecule, fondaparinux analogue, has not been synthesized.

A1) We definitely agree with the reviewer that it is not a complete work; however, the remaining steps are quite routine in the state of art and hence, we have not repeated. As the final deprotection and sulfation for fondaparinux is well established by other practitioners (Dey, S.; Wong, C.-H. Chem. Sci. 2018, 9, 6685-6691; Carbohydrate Res. 1987, 167, 67-75; Angew. Chem. Int. Ed. 2014, 53, 9876-9879). In this manuscript, we have showed the power of gold-catalysis for the synthesis of a pharmaceutically relevant pentasaccharide and while doing so, we synthesized the very important regioselective deprotections hitherto that was noted to be difficult. The partially deprotected regioisomeric hydroxyls will enable regioselective sulfation to ensue studies of their biological properties. That is why we have given the title as fondaparinux pentasaccharide instead of the just fondaparinux. Through this manuscript, our main intention was to show that gold-catalyzed glycosidation is excellent for the synthesis for the complex molecules as well.

Q2) Only a fondaparinux analogue could be obtained from the total protective pentasaccharide. In order to get the beta-isomer of GlcA glycosidation products, a Bz, Ac or other acyl group must be assembled on the 2-OH because of the strategy defects, however, the 2-OH on GlcA was not sulfated. So insightfully, the catalytic strategy is not suitable for the synthesis of fondaparinux.

A2) We sincerely thank reviewer for identifying this problem.

While synthesizing the Glc β (1 \rightarrow 4)GlcN3, we have encountered with number of observations that enticed us retention of the C-2 benzoate for obtaining the much desired 1,2-*trans* selectivity. Here is the explanation:

Compound **SI.1** was subjected to the Zemplén deacylation conditions, continued to protect the benzylidene using benzylidene dimethyl acetal, CSA to notice that the benzylidene as well as the benzoate; NMR and mass spectral analysis of the resultant compound **SI.2** showed that the benzoate is at the C-2 position only. Hence, it was envisioned that the C-2-OBz survives Zemplén deacylation conditions thereby facilitating the cleavage of other acylates in the presence of C-2-OBz *en route* to the synthesis of the desired pentasaccharide. This particular observation saved at least three steps for the overall synthesis.

If the reviewer feels that this is an important aspect in the synthesis, then, we can move it to the main manuscript as well. Once again, we apologise and thank reviewer for bringing this up for the discussion.

SI.1

Calc. $[C_{67}H_{69}N_3O_{14}Si+Na]^+$ = 1190.4447
 Observed = 1190.4437

1. NaOMe (0.3 equiv. to excess)
 $CH_2Cl_2:MeOH$ (1:1), 25 °C
 30 min to 8 h, 85%
2. $PhCH(OMe)_2$, CSA
 DMF , 4 h, 25 °C, 71%

SI.2

Calc. $[C_{60}H_{65}N_3O_{12}Si+Na]^+$ = 1070.4235
 Observed = 1070.4224

1H NMR ($CDCl_3$, 400.31 MHz) of Compound SI.2

^{13}C NMR ($CDCl_3$, 100.12 MHz) of Compound SI.2

DEPT NMR ($CDCl_3$, 100.12 MHz) of Compound SI.2

Q3) And a Me group was installed on the reducing end rather than an All group on fondaparinux molecule.

A3) We are aware that there will be a methyl group in Arixtra. Methyl group can be installed by subjecting it to two additional steps *viz.* hydrolysis of the allyl ether, conversion to the carbonate donor and then, performing the glycosidation with methanol as the aglycon. Alternatively, one can also convert the hemiacetal after the PdCl₂ reaction to the Hashimoto's protocol (Ag₂O/Mel) {e.g. *Chem. Sci.* **2018**, *9*, 6685-6691}. So, this route enables that methyl glycoside as well. We have placed the allyl group for the following reasons:

- 1) Final deprotection and sulfation to prepare therapeutically relevant Fondaparinux will automatically convert allyl moiety to the *n*-propyl moiety while performing the hydrogenolysis of benzyl ethers.
- 2) Allyl moiety can be hydrolysed and one can prepare the carbonate donor again for further glycosidations to elongate the chain length for the heparin oligosaccharide synthesis plan OR it can be glycosylated with other biological relevant nucleophiles (ref. Mishra, B.; Neralkar, M.; Hotha, S. *Angew. Chem. Int. Ed.* **2016**, *55*, 7786-7791).
- 3) Allyl moiety can be subjected to a variety of ligation reactions such as cross metathesis or thiol-ene click reaction etc. for conjugating bioprobes.

Q4) The Troc of the building block of H was not a suitable protecting group, because of several unnecessary steps were needed. And in some other reported papers, the Cbz was selected as an appropriate protecting group, because it can be removed by H₂ along with Bn and N₃.

A4) The Troc-group was kept for preparing the benzylidene and then quickly converted to the N₃ as the reviewer noted. Cbz was not selected because the deprotection of Cbz is necessary for converting to the N₃ and during the process, benzylidene will also be susceptible. Keeping the orthogonality of the protecting groups and the overall plan of synthesis, we have made use of the Troc-protecting group on the C-2-NH₂ group rather than the Cbz. Also, the presence of Troc moiety gave us the additional advantage while purifying the building block as the ΔR_f was larger.

Q5) In order to using the Au/Ag catalytic method, the All must be removed and subsequently the leaving group must be installed on the donor, and several unnecessary steps were needed. So it is not a simple method. Furthermore, the Au and Ag are precious metals, so it is not economical for scalable synthesis of Fonda.

A5) We agree that Au and Ag are precious metals; however, they are pretty cheap as catalysts. For example: Price of 500 mg of Au-phosphite catalyst is INR 5138 and that of 1 g of Grubbs 1st Generation catalyst costs INR 10,478 which means that it is INR 5239. So, the price of the catalyst is not a problem at all. It is only the

public perception that gold- and silver- are expensive. Also, it is dependent on the significance of the molecule as well. Herein, we are activating a very stable glycosyl donor. The reactions are at least 40% more efficient and clean. NO offensive by products etc.

Regarding the use of allyl group:

Placing the allyl group increases the process efficiency as it can be activated latently and will not effect other reactions and in addition, it can be regioselectively cleaved off whenever it is required. Fittingly, allyl is quite suitable as:

- a) As allyl can be installed at an early stage that diminishes number of steps for orthogonal protections, useful to synthesize number of diverse orthogonal building blocks from a single intermediate.
- b) Allyl is quite stable towards other protecting group interconversions.
- c) Regioselectively cleavable even in the presence of acid sensitive groups such as PMB, Benzylidene, Silyl protections etc.

Q6) The title in Supporting Information is different from the title in article.

A6) Thank you very much for the suggestion. In view of the comments from the reviewers, we have modified the title and the new title has been placed in both the files.

Q7) Line 107, the 18a need to be changed to 18a, line 163, the 30a-30e need to be changed to 30a-30e. and please carefully check the similar mistakes also in this paper.

A7) Thank you very much for the suggestion. We have corrected.

Q8) In the PDF format, almost all the "oC" was changed to " ".

A7) Thank you very much. It is a problem while converting it from Word to the Pdf.

Q9) Recently, two papers about the synthesis of Idraparinux and Arixtra have been reported, so these two papers needed to be cited in the article: Journal of the American Chemical Society, 2019, 141, 26, 10309-10314 and Org. Chem. Front., 2019, 6, 3116-3120.

A9) Thank you very much for the suggestion. We have added these references also now.

Q10) Line 107, the 18a need to be changed to 18a, line 163, the 30a-30e need to be changed to 30a-30e. and please carefully check the similar mistakes also in this paper.

A10) Thank you very much for the suggestion. We have corrected.

Reviewer 2:

Thank you very much for observing our manuscript to be suitable for publishing in Communications Chemistry.

General: This manuscript describes the formal synthesis of the anticoagulant Fondaparinux pentasaccharide by employing catalytic promoting system for all glycosidations. The synthesis of heparin and heparan sulfate remains a challenge in carbohydrate chemistry. The low degree of the accessibility to monosaccharides involved in them such as glucuronic acid and iduronic acid and the difficulty of optimizing pairs of orthogonal protecting groups to attain efficient assembly of oligosaccharide sequences in a stereoselective manner has defied many challenges of the chemical synthesis of heparin and heparan sulfate. Therefore,

anti coagulant Fondaparinux that is the target molecule in this study has been chemically synthesized by few groups despite the strong demand for its clinical use.

In the present study, the authors have succeeded in highly stereoselective and efficient synthesis of orthogonally protected Fondapaninix pentasaccharides for the first time, which will allow for structure-activity relationship study of Fondapaninix. The authors showcased the efficacy of the silver assisted gold catalyzed glycosidation developed by the author's group, which provided high yields and stereoselectivity in all the glycosidation reactions, almost of which exceed the yields reported in the literatures. The iduronic acid synthesis via the chelation assisted Grignard addition strategy is also interesting. This method will be very useful for large scale preparation. It is overwhelming that 120 mg of oligosaccharide intermediate was synthesized by the authors. I believe this method will fulfill the demand for large scale synthesis of the Fondapaninix and its analogs. The experimental procedures are detailed and useful for others to reproduce the work.

Answer: Thank you very much for very encouraging words.

Q1. p7, line 145: the conversion into 5 does not seem "uneventfully occurred" due to the middle conversion yield (55%). Additional statement about the byproducts should be involved in the main text.

A1) The confusing statement has been modified in the revised manuscript. The starting materials remained as such which were isolated from the isopropylidene and subjected again to the same reaction so that overall the reaction could give much better yield. We mentioned the same in the revised manuscript. Thank you very much for noting it.

Q2. Scheme 2, reaction j: The reason of the use of dichloroacetic acid should be mentioned in the main text.

A2. Thank you very much for query. Initially, we have conducted the reaction in trifluoroacetic acid ($pK_a = 0.0$) and trichloroacetic acid ($pK_a = 0.77$) to observe that the TBDMS also cleaved off; then, we tried the dichloroacetic acid ($pK_a = 1.25$) which has given us the cleavage of the desired isopropylidene without affecting the TBDMS-ether. { pK_a values are taken from (<https://www2.chemistry.msu.edu/faculty/reusch/virttxtjml/acidity2.htm>)}

Q3. 33e (6-O-Lev-GlcN3) will also fit with the reciprocal strategy like 31a (6-O-Bz-GlcN3), which in addition makes selective deprotection enable in pentasaccharide. Is there any reason why 6-O-Lev-GlcN3 acceptor was not used in the glycosidation examinations in Table 2?

A3. One can also do such kind of reaction; however, one has to be careful about the ease of purification. In addition, we have not tried that because, the saponification of benzoates and acrylates is to be performed for sulfation at the C-6 position. Based on our experience with synthesis of mycobacterial glycosides, the Lev esters are not very easy to deprotect under standard Zemplen conditions. So, it will add an additional step and hence, we have not screened it. Thank you very much for the observation.

Q4. p 14, line 261: the statement 'allyl moiety of the disaccharide 39i was smoothly hydrolyzed....' is not correct. This reaction is not hydrolysis.

A4. Thank you very much for the suggestion. We have now changed it to cleaved off in the revised manuscript.

Q5. The subscripted numbers with R (R1, R2) in the schemes should be superscripted.

A5. Thank you very much for the suggestion. We have modified subscripts to superscripts in the revised manuscript.

Reviewer 3:

Thank you very much for your kind attention about our manuscript and constructive reviewing.

General 1: Hotha and co-workers describe their silver-assisted gold catalysis for the anticoagulant fondaparinux pentasaccharide synthesis utilizing a 3 + 1 + 1 and 3 + 2 strategy. To date, several scientific groups have made the exact same target pentasaccharide molecule, more or less, using their own unique in-house chemistry. The synthetic routes applied to obtain the monosugar units are well-known as the authors have reported these in the bibliographic references (see references 22-27). The manuscript, itself, is well-written. Two iduronates were prepared, cleverly, and this author has to be given credit for that. May derivatives were also made for 30, 31, 33, 36, 37, 38 and 39.

Answer: Thank you very much for the encouragement.

General 2: This reviewer, however, understands that the authors did lot of work on donor/acceptor selectivity studies to get suitably amounts of product after glycosylation. The authors found that 39i is the best disaccharide to further elaborate their chemistry with. However, it failed to give the desired trisaccharide. After further standardization, the team was then able to show that 44 was the disaccharide to proceed onward with. From 44, the authors prepared trisaccharide 47 which was split into two parts to ultimately achieve pentasaccharide 50 which was again split into two parts to get partially deprotected pentasaccharide 53.

Answer: Donor-acceptor selectivity studies have guided us to come with the right combination for preparing the desired trisaccharide 47 with good control over stereoselectivity as well as the polarity differences for easy purification of the trisaccharide and intermediate steps.

General 3: The authors conclude that these regioisomeric hydroxyls will be highly useful for the sulfation to study their biological properties. However, this reviewer believes to get final deprotection with suitable sulfonated is very necessary to get high standard publication. All the shown spectral quality is very good but 2D spectra is necessary, once again, for at least from trisaccharide unit. So, the authors are being requested to provide that spectra and as well as show the complete synthesis of Fondaparinux (1) as it is shown in Scheme 1.

Answer: We definitely agree with the reviewer that it is not a complete work; however, the remaining steps are quite routine in the state of art and hence, we have not repeated. As the final deprotection and sulfation for fondaparinux is well established by other practitioners (e.g. Dey, S.; Wong, C.-H. Chem. Sci. 2018, 9, 6685-6691; Carbohydrate Res. 1987, 167, 67-75; Angew. Chem. Int. Ed. 2014, 53, 9876-9879). In this manuscript, we have showed the power of gold-catalysis for the synthesis of a pharmaceutically relevant pentasaccharide and while doing so, we synthesized the very important regioselective deprotections hitherto noted to be difficult. The partially deprotected regioisomeric hydroxyls will enable regioselective sulfation to ensue studies of their biological properties. That is why we have given the title as fondaparinux pentasaccharide instead of the just fondaparinux. Through this manuscript, our main intention was to show that gold-catalyzed glycosidation is excellent for the synthesis for the complex molecules as well.

We believe that this route is simple compared to what is existing out there in the literature for the following reasons: (i) the route stands on the stable glycosyl donor chemistry; (ii) catalytic glycosidations that are high yielding; (iii) the purifications are easier due to the choice of protecting groups and the polarity differences; (iv) all the glycosidations occur in a stereoselective fashion thereby reducing the tedious and lengthy column chromatographies; (v) finally, one of the first catalytic routes for the fondaparinux pentasaccharide. (vi) glycosidations can be performed easily.

General 4: This reviewer does not recommend publication in Journal of communications chemistry based on the manuscripts current state, but with a lengthy time-frame to make the paper more mature, the journal should reconsider reviewing it.

Answer: Thank you very much. Soon after receiving your comments, the Corona pandemic set in and laboratories got closed for more than 6 months. In the meantime, some of the intermediates have developed some impurities and some have degraded and that required us to repurify/resynthesize. We have now acquired the 2D spectra for some compounds and given that data also in the revised supporting information. We have changed the title as well keeping in view of the facts. We do hope that this manuscript will now become suitable for publication and receives your positive feedback.

Q1. Spectral data needs to be reported with specific numbers. Please mention specific numbers (ppm shifts) for ALL 1H or 13C belonging to each sugar.

A1. Most of the signals are overlapping and hence, assignment of chemical shifts to specific sugars is very challenging. However, the anomeric region comes separately and hence, assignment to that region is possible and we have done that wherever it is possible.

Q2. Add this as reference: Page 4 line 75: Preactivation-based, iterative one-pot synthesis of anticoagulant pentasaccharide fondaparinux sodium [Org. Chem. Front., 2019,6, 3116-3120]

A2. Thank you very much. We have added this also in the revised manuscript.

Q3. Page 7, line 148: Is that 27 reference correct for this sentence? Or should it be 28? We will correct it in the revised manuscript

A3. Thank you very much for the suggestion. Yes, the reviewer is right that the number should be 28. We have inadvertently placed 27. We have corrected in the revised manuscript.

Q4. Page 3, Scheme 1. In retrosynthetic scheme use 'number 2' for protected only not for the deprotected pentasaccharide.

A4. Thank you very much for the suggestion. We have corrected in the revised manuscript.

Q5. From 17a to 18a conversion: Reviewer would like to see the plausible mechanistic explanation. Furthermore, provide a possible reason for not forming product in the case of benzyl protected C-3 position.

A5. It is known that RuCl_3 in the presence of NaIO_4 results in the formation of RuO_4 which is a very powerful oxidant (See ref: Naota, T.; Takaya, H. Murahashi, S.-I., *Chem. Rev.* **1998**, *98*, 2599-2660). The current protocol of using phenyl group as a surrogate for carboxylic acid was first reported by Nunez and Martin in 1990 (see: Nunez, M. T.; Martin, V. S. *J. Org. Chem.* **1990**, *55*, 1928-1932). Regarding non-conversion of compound **17b** to **18b**: The phenyl group present as benzyloxy moiety at the C-3 position in **17b**. We got a complex mixture of products from which the compound **18b** could not be isolated successfully as the phenyl group present in the C3-OBn can also get oxidized to give many products. Nunez and Martin JOC paper is cited in the revised manuscript.

Q6. In page 15: line 288-294 and page17, line 304-308: Which route is better to achieve the synthesis of compound 50? The authors need to conclude that one.

A6. It is better to follow the 3+2 route than the 3+1+1 route as the former is more convergent. The 3+1+1 route will be useful for making more derivatives of the pentasaccharides. A statement with regards to this has been added in the revised manuscript.

Q7. All the shown spectral quality is very good but 2D spectra is necessary

A7. Thank you very much for the suggestion. Soon after receiving, me and my co-workers have pondered over and started the work that is required to submit the revision. However, our institution is closed due to the Covid-19 pandemic and slowly the organization is allowing now. Some of the intermediates have survived and some needed some exhaustive purification. So, we purified them again and obtained 2D-NMR spectra and interpreted them. The conclusions and other things did not change due to this characterization; though we got more insight into the overall structural homogeneity. We have provided 2D-NMR spectral copies in the revised SI. Once again, thank you very much.

Overall, we thank honourable reviewers for constructive reviewing and recommendations.

REVIEWERS' COMMENTS:

Reviewer #1 (Remarks to the Author):

[Editorial note: This reviewer provided no further comments to the authors.]

Reviewer #2 (Remarks to the Author):

Hotha and his co-authors have revised the original manuscript describing the formal synthesis of anticoagulant Fondaparinux pentasaccharide by considering all the comments of this reviewer point by point. All the questions given by this reviewer have been clearly answered by the authors. I am happy to recommend the revised manuscript for publication in Communication Chemistry as it stands.

Reviewer #3 (Remarks to the Author):

1) This reviewer recommends that the authors include 2D spectrum from trisaccharide (46) to pentasaccharide (53) not only 46 and 48.

2) Point-wise response to Reviewers' Observations: Reviewer 3. Q1: Author is requested to mark (may be highlighting) the changes in supporting, it will be helpful for reviewer to understand the changes quickly. Author referenced *Angew. Chem. Int. Ed.* 2014, 53, 9876-9879. The authors might want follow the style for representing NMR data for their current manuscript.

3) Draw the structure for NMR data representation. For example, if you are presenting NMR data for 44 then structure of 44 should be placed next to it. This helps the reader to pick out the structure quickly providing a better experience.